# Cadence Feedback and Video-Based Engagement Improves Motivation and Performance during Pedalling in Stroke Patients

Mukesh Soni [1] , Tissa Wijeratne [2,3,4] and David Ackland [1,*]

[1] Department of Biomedical Engineering, University of Melbourne, Parkville, VIC 3010, Australia; mukesh.soni@unimelb.edu.au

[2] Department of Medicine and Neurology, AIMSS, Melbourne Medical School, University of Melbourne and Western Health, Sunshine Hospital, St Albans, VIC 3021, Australia; tissa.wijeratne@wh.org.au

[3] Department of Psychology & Counselling, School of Psychology & Public Health, La Trobe University, Bundoora, VIC 3086, Australia

[4] Department of Medicine, Faculty of Medicine, University of Rajarata, Saliyapura, Anuradhapura 50300, Sri Lanka

[*] Correspondence: dackland@unimelb.edu.au; Tel.: +61-4-07-823-190

**Abstract:** Video and music as a dissociative attention stimulus during exercise is known to distract from the discomfort of physical exertion and improve exercise adherence; however, the influence of video-based feedback and engagement during pedalling on the performance and motivation of pedalling in stroke patients is poorly understood. The aim of this study was to employ a novel video-based engagement paradigm for pedalling in stroke patients and evaluate its capacity to influence the cadence control, physiological output, and perceived motivation and enjoyment. Thirteen stroke patients were recruited with low-to-moderate lower-limb disability (mean age: 64.0 yrs.). A reference group of 18 healthy young adult subjects (mean age: 27.7 yrs.) was also recruited to assess the broad applicability of the techniques to a contrasting non-pathological cohort. The participants pedalled at a slow (60 RPM) and fast (100 RPM) target speed with constant resistance in 15 min pedalling bouts that included (i) baseline pedalling with real-time visual feedback of cadence deviation from the target provided only in the first 20 s (ii) real-time visual feedback of cadence data over the entire pedalling session, and (iii) real-time engagement to maintain the playback rate of a prerecorded video by pedalling at the target speed. During low speed pedalling, stroke patients demonstrated significantly smaller absolute cadence deviation during pedalling with feedback (mean difference: 1.8 RPM, $p = 0.014$) and video-based engagement (mean difference: 2.4 RPM, $p = 0.006$) compared to the baseline pedalling. For the healthy adults, feedback and video-based engagement reduced cadence deviation significantly at all speeds ($p < 0.05$). All but one stroke patient either enjoyed or really enjoyed the video engagement during pedalling and felt motivated to undertake this form of exercise in therapy in the future. This proof-of-concept study showed that feedback and video-based engagement may improve the targeted pedalling performance in stroke patients, and by helping dissociate subjects from physical cues associated with fatigue, may ultimately improve exercise motivation and compliance.

**Keywords:** rehabilitation; therapy; exercise performance; cadence control

## 1. Introduction

Stroke is the second-leading global cause of death behind heart disease, accounting for 11% of total deaths worldwide [1]. Approximately 50% of stroke survivors suffer reduced mobility due to paralysis in a lower limb [1]. Individuals with post-stroke hemiparesis, who exhibit asymmetrical muscle activation, reduced motor control, combined with metabolic and structural changes within the skeletal muscle, also tend to have a decline in balance

and ambulation capacity [2]. Pedalling-based therapy, which generates cyclic and symmetric power output without the requirement of balance maintenance, has been shown to improve patient aerobic capacity [3,4], ambulatory balance, and motor performance in stroke patients [5,6]. It can be performed in the home while seated and without lower limb weight bearing, and is considered a safe alternative to walking in subjects with postural instability and fall risk [3,7].

The repetitive nature of pedalling-based exercise therapy on a stationary bicycle has been associated with poor motivation, low compliance, and lack of perceived self-efficacy, which are key barriers to the use of pedalling in rehabilitation [8]. The use of real-time measurement and display of exercise performance during cycling, including power, speed, speed variability, distance, and duration of the exercise, has been shown to increase exercise motivation and compliance in stroke patients [7,9–11]. Furthermore, real-time visual feedback on the work completed by the lower limbs during pedalling has been attributed to reduced limb-to-limb pedalling imbalance and increased pedalling speed in chronic stroke patients, resulting in a higher gait speed and reduced gait asymmetry [12]. At present, however, the influence of real-time biofeedback of cadence control at different speeds, which may ultimately influence the effectiveness of pedalling-based therapy, remains poorly understood.

The use of video and music during exercise therapy has been shown to distract from the physical discomfort of exertion [13–15], acting as a dissociative attentional stimuli, which may lower the perceived exertion and improve exercise performance [16–18]. Robergs et al. (1998) demonstrated that pedalling while watching a prerecorded video in healthy individuals contributed to higher peak pedalling speeds for a given blood lactate level compared to pedalling without video interaction [19], while MacRae et al. (2003), in a cohort of healthy women, showed that video and music stimulus while pedalling improved peak speed and the overall distance pedalled [20]. While there is evidence of the role of feedback and video engagement on cardiovascular performance in healthy individuals, including eliciting peak performance and extending time to exhaustion [21], the ability for video engagement to improve exercise performance and motivation in stroke patients has not been demonstrated to date. Stroke survivors that present with motor impairments have a characteristically lower ventilatory threshold and reduced muscle strength [22,23], which may compromise capacity to achieve targeted pedalling goals, even in those affected by the most mild lower limb functional impairments [24]. Cognitive impairment and attention deficits may pose a further challenge to stroke survivors attempting to leverage the benefits of video-based feedback and engagement during exercise [25,26].

The objective of this study was to evaluate the influence of a novel video-based engagement regime for pedalling in stroke patients, one that required subjects to actively pedal in order to control the play rate of a prerecorded video. The specific aims were: first, to evaluate the capacity of this exercise regime to influence cadence control and the perceived motivation and enjoyment in both stroke patients and healthy controls during pedalling at different speeds; and second, to compare these findings to the cases of pedalling with cadence performance feedback only, as well as pedalling with no feedback or engagement. We hypothesized that video-based engagement during pedalling and cadence feedback would improve pedalling performance and enjoyment by providing a form of dissociation from bodily cues associated with fatigue.

## 2. Materials and Methods

### 2.1. Subject Recruitment

Thirteen stroke patients with mild to moderate lower limb disability (Modified Ranking Scale: 2–3) were recruited to this study (mean age: 64.0 years, range: 25–78 years; mean weight: 77.5 kg, range: 53–127 kg; mean height: 167.9 cm, range: 150–177 cm, 10 male: 3 female). Patients had no history of joint pain, dysfunction, or previous lower-limb surgery and undertook varying levels of physical activity in their lives prior to the stroke event (no exercise/activity: 2 participants, low/occasion: 3 participants, moderate: 3 partici-

pants, regular/daily: 5 participants). Patient cognition function was assessed based on the capacity to: (i) understand the nature and purpose of the study, (ii) understand the risks and benefits of the study, (iii) understand their right to participate and withdraw, and (iv) provide informed consent to participate in the study. A group of 18 healthy young adults were also recruited in order to assess the general applicability of the proposed methods to a contrasting, non-pathological, and able-bodied cohort (mean age: 27.7 years, range: 25–41 years; mean weight: 61.6 kg, range: 40–86 kg; mean height: 165.3 cm, range: 151–182 cm, 8 male: 10 female). These healthy participants were physically active and most exercised regularly (no exercise/activity: 2 participants, low/occasion: 2 participants, moderate: 9 participants, regular/daily: 5 participants). All participants were required to have the capacity to read and communicate orally in English. Ethical approval for this study was obtained, and all participants provided written informed consent.

### 2.2. Testing Protocol

A commercially available stationary pedalling apparatus (BL3700, Novis Healthcare, Lane Cove West, NSW, Australia) was modified to provide real-time video engagement during pedalling as well as feedback on pedalling performance (Figure 1). The test setup comprised an adjustable seat, pedals, an exercise data recorder, and a computer with a display. To achieve optimal knee flexion and comfort during pedalling, the seat and pedal crank positions were adjusted to each subject's lower limb length and function. Speed and cadence were measured by instrumenting the pedalling apparatus crank using a magnet and hall-effects cadence sensor with a microcontroller board (Mega2560, Arduino, Ivrea, Italy). These data were wirelessly transmitted to a PC, which provided real-time data during pedalling via the display screen (Visual Studio 2017, Microsoft, Redmond, WA, USA), including pedalling speed and duration, target speed, and deviation from target speed data (Figure 2). Real-time video engagement was facilitated by displaying a prerecorded video of the participant's choice from a collection offered by the researchers. The video playback rate was synchronised to a target pedalling speed, and to maintain the video playback rate, participants were required to maintain their pedalling at the target speed. An increase or decrease in cadence from the target speed by 5 rotations per minute (RPM) or more from the target resulted in the video playback moving in fast-forward or slow motion, respectively.

Each participant was first given one minute to familiarise themselves with the testing apparatus. They then undertook three randomly allocated 15-min sessions of pedalling targeting a set low cadence (60 RPM) which included (i) baseline pedalling without feedback or engagement and cadence information provided only in the first 20 s in order to establish target cadence (ii) real-time visual feedback of cadence data for the entire session, including pedalling duration, pedalling cadence, and cadence deviation from the target, and (iii) real-time engagement to maintain the playback rate of a prerecorded video by pedalling at the target speed. These testing conditions were subsequently repeated at a high target cadence (100 RPM). Thirty-second rest periods were provided after every 5 min of pedalling.

On completion of the testing, participants completed a self-reported feedback survey that evaluated (i) perceptions of differences between pedalling with visual feedback and video-based engagement compared to the baseline pedalling; (ii) overall experience of pedalling with feedback or engagement, from enjoyment to hindrance; (iii) use of feedback during pedalling for future exercise therapy; and (iv) potential for feedback during pedalling to influence exercise motivation (Supplementary Materials). The survey was undertaken twice, once for pedalling sessions that provided cadence data feedback and a second time for sessions that employed real-time video-based engagement. Before providing responses, the survey questions were discussed with each participant to ensure consistency in question interpretation and reporting.

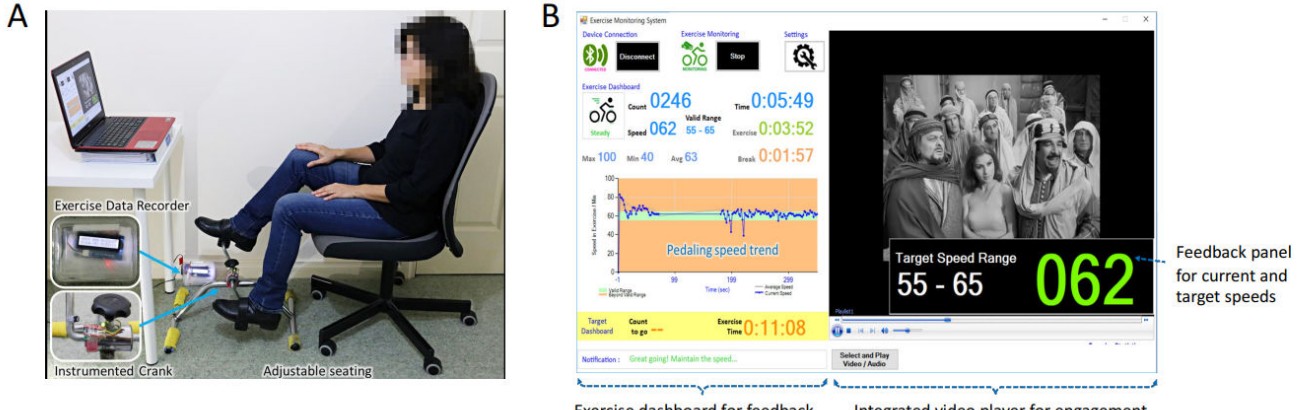

**Figure 1.** Experimental setup of a custom-designed apparatus used for pedalling trials (**A**) and software graphic user interface (GUI) on a PC used for feedback and engagement (**B**). The pedalling apparatus comprised an adjustable computer seat, a pair of pedals, a crank instrumented with a cadence sensor, a data recorder to measure pedalling cadence, and a customised software suite running from the computer. The software GUI displayed exercise performance feedback on a dashboard, while an integrated video/music player was also displayed to provide engagement during exercise.

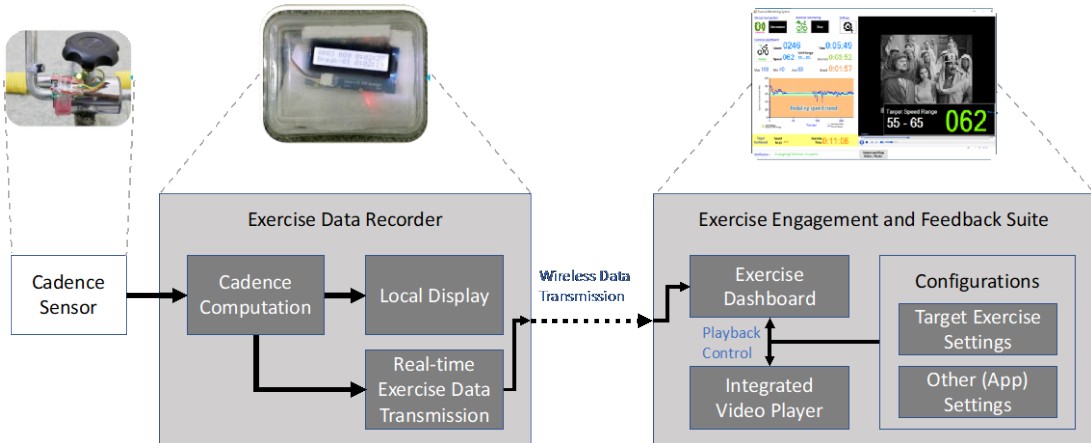

**Figure 2.** Data workflow for the pedalling apparatus, which integrated cadence data recorded from a cadence sensor that was output to a local display and transmitted wirelessly to a PC via an exercise data recorder. The software-based exercise engagement and feedback suite featured exercise performance data visible on a dashboard, an integrated video player used for engagement, as well as a configurations menu used for programming the task settings.

*2.3. Data Analysis*

The stroke cohort size was selected as a sample of convenience, while the young adult group sample was chosen on the basis of a previously published study [27]. The effect of intervention type (pedalling with cadence feedback, pedalling with video-based engagement, and the baseline pedalling condition) on cadence while pedalling at slow and fast cadences was evaluated using a two-way Analysis of Variance (ANOVA). Mean-cadence deviation and the Coefficient of Variation (CV) in mean cadence were the dependent variables in this analysis. Interactions between the independent variables were also assessed with the ANOVA. Post hoc testing was undertaken using a Games-Howell test for groups with unequal variances. Levene's test for homogeneity of variances was performed, and the data normality was verified. The standard deviation was evaluated and used to quantify data dispersion. The level of significance was defined at $p \leq 0.05$.

## 3. Results

### 3.1. Cadence Control

During low-speed pedalling, stroke patients demonstrated significantly smaller absolute cadence deviation while pedalling with feedback (mean difference = −1.8 RPM, $p = 0.014$) and engagement (mean difference = −2.4 RPM, $p = 0.006$) compared to the baseline pedalling (Table 1), with no change in cadence deviation observed during high-speed pedalling. For the healthy adult subjects, cadence deviation was significantly lower with feedback (mean difference = −3.0 RPM, $p = 0.028$) and engagement (mean difference = −3.4 RPM, $p = 0.010$) compared to that during baseline pedalling at low speed. A significantly lower cadence deviation was also observed during pedalling with feedback (mean difference = −3.1 RPM, $p = 0.002$) and engagement (mean difference = −2.2 RPM, $p = 0.007$) at high speed ($p < 0.05$). Greater variance in cadence around the target was observed in stroke patients compared to the healthy adults. Specifically, the COV of the absolute speed deviation was significantly higher in stroke patients compared to that in healthy adults during pedalling at low speed with feedback (mean difference = 1.7%, $p = 0.013$) and engagement (mean difference = 1.7%, $p = 0.008$), as well as during high-speed pedalling with feedback (mean difference = 1.0%, $p = 0.016$) and engagement (mean difference = 1.0%, $p = 0.018$).

### 3.2. Self-Reported Outcome Measures

The followup survey showed that 11 stroke patients (84.6%) and 17 healthy young adult subjects (94.4%) observed a difference in pedalling with feedback or engagement relative to pedalling without (Table 2). Of the stroke patients, two subjects (15.4%), eight subjects (61.5%), one subject (7.7%), and two subjects (15.4%) found feedback during pedalling to be really enjoyable, enjoyable, neutral or slightly troublesome, respectively. When engagement was provided, the number of stroke patients that enjoyed and really enjoyed pedalling increased to three (23.1%) and nine (69.2%), respectively. Eleven stroke subjects (84.6%) believed both feedback and engagement to be beneficial to their exercise routine, and 12 stroke subjects (92.3%) felt that feedback and engagement would help to motivate them to adhere to pedalling-based exercise. All stroke participants indicated they would choose to pedal with engagement in the future if given the opportunity, and 11 patients (84.6%) would elect to pedal with feedback.

Of the healthy subjects, 16 (88.9%) believed that pedalling with feedback and engagement would assist them with their exercise, while 16 subjects (88.9%) and 17 subjects (94.4%) believed that feedback and engagement would help motivate them to adhere to pedalling exercise, respectively. Fourteen healthy adult subjects (77.8%) would perform pedalling with video-based engagement if given the opportunity, and 15 subjects (83.3%) would elect to undertake pedalling with feedback. Written feedback from the survey indicated that the majority of stroke patients (81.8%) and healthy adults (82.4%) found feedback helpful or useful in their performance tracking, while most stroke patients (92.3%) and healthy adults (70.6%) found engagement both relaxing and appealing as an exercise therapy.

**Table 1.** Results of the three-way analysis of variance (ANOVA) for the cadence control parameters during pedalling at low and high speeds. The dependent variables were absolute cadence deviation from the target cadence (RPM) and the coefficients of variation (%) of the cadence deviation. The independent variables were intervention type (baseline pedalling, pedalling with performance feedback, and pedalling with video-based engagement), low and high pedalling speed (60 RPM and 100 RPM, respectively), and participant type (healthy adults or stroke patient). For each variable, mean and standard deviation (SD) are reported. Significant differences in intervention type (Int), pedalling cadence (Cad), subject type (Sub), and their interactions are given, specifically, Int*Cad, Int*Sub, Cad*Sub, and Int*Cad*Sub. See Supplementary Materials for effect size data.

| | | Healthy Adults | | | | | | Stroke Patients | | | | | | Individual Effect | | | Interaction Effect | | | |
| | | Baseline | | Feedback | | Engagement | | Baseline | | Feedback | | Engagement | | | | | | | | |
| | Cadence | Mean | SD | Mean | SD | Mean | SD | Mean | SD | Mean | SD | Mean | SD | Int. | Cad. | Sub. | Int*Cad | Int*Sub | Cad*Sub | Int*Cad*Sub |
|---|---|---|---|---|---|---|---|---|---|---|---|---|---|---|---|---|---|---|---|---|
| Absolute cadence deviation | Low | 6.1 | 4.1 | 3.1 | 2.0 | 2.7 | 0.9 | 7.0 | 4.4 | 5.2 | 4.0 | 4.6 | 2.8 | $p < 0.001$ | $p = 0.355$ | $p = 0.108$ | $p = 0.140$ | $p = 0.007$ | $p = 0.606$ | $p = 0.532$ |
| | High | 6.8 | 3.3 | 3.7 | 2.0 | 4.6 | 4.6 | 5.8 | 3.7 | 6.0 | 3.9 | 5.9 | 3.2 | | | | | | | |
| Cofficient of Variation | Low | 5.0 | 1.1 | 4.2 | 1.2 | 3.9 | 1.0 | 6.0 | 2.0 | 5.9 | 2.0 | 5.6 | 1.9 | $p < 0.001$ | $p < 0.001$ | $p = 0.006$ | $p = 0.459$ | $p = 0.008$ | $p = 0.098$ | $p = 0.995$ |
| | High | 3.7 | 1.1 | 2.9 | 1.0 | 2.9 | 0.7 | 4.0 | 1.6 | 3.9 | 1.1 | 3.9 | 1.3 | | | | | | | |

**Table 2.** Self-reported outcome survey, including results for both stroke patients and healthy young adult participants.

| Question | | Healthy Adults (n = 18) | Stroke Patients (n = 13) |
|---|---|---|---|
| Did you observe a difference between excerise sessions when you received feedback compared to when you didn't? | Yes | 17 (94.4%) | 11 (84.6%) |
| | No | 1 (5.6%) | 2 (15.4%) |
| | I don't know | 0 (0%) | 0 (0%) |
| How did you find your experience of pedaling with performance feedback? | 5. Really enjoyed | 4 (22.2%) | 2 (15.4%) |
| | 4. Enjoyed | 10 (55.6%) | 8 (61.5%) |
| | 3. Neutral | 3 (16.7%) | 1 (7.7%) |
| | 2. Slightly troublesome | 1 (5.6%) | 2 (15.4%) |
| | 1. A hindrance | 0 (0%) | 0 (0%) |
| How did you find your experience of pedaling with video-based engagement? | 5. Really enjoyed | 8 (44.4%) | 3 (23.1%) |
| | 4. Enjoyed | 7 (38.9%) | 9 (69.2%) |
| | 3. Neutral | 2 (11.1%) | 1 (7.7%) |
| | 2. Slightly troublesome | 1 (5.6%) | 0 (0%) |
| | 1. A hindrance | 0 (0%) | 0 (0%) |
| Would you perform pedaling exercise with performance feedback in the future if given the opportunity? | Yes | 15 (83.3%) | 11 (84.6%) |
| | No | 0 (0%) | 1 (7.7%) |
| | I don't know | 3 (16.7%) | 1 (7.7%) |
| Would you perform pedaling exercise with video-based engagement in the future if given the opportunity? | Yes | 14 (77.8%) | 13 (100.0%) |
| | No | 1 (5.6%) | 0 (0%) |
| | I don't know | 3 (16.7%) | 0 (0%) |
| Do you believe performance feedback during pedalling is of benefit to your exercise? | Yes | 16 (88.9%) | 11 (84.6%) |
| | No | 2 (11.1%) | 2 (15.4%) |
| | I don't know | 0 (0%) | 0 (0%) |
| Do you believe video-based engagement during pedaling is of benefit to your exercise? | Yes | 16 (88.9%) | 11 (84.6%) |
| | No | 2 (11.1%) | 2 (15.4%) |
| | I don't know | 0 (0%) | 0 (0%) |
| Would performance feedback during pedaling encourage you to exercise more? | Yes | 16 (88.9%) | 12 (92.3%) |
| | No | 2 (11.1%) | 2 (15.4%) |
| | I don't know | 0 (0%) | 0 (0%) |
| Would video-based engagement during pedaling encourage you to exercise more? | Yes | 17 (94.4%) | 12 (92.3%) |
| | No | 2 (11.1%) | 2 (15.4%) |
| | I don't know | 0 (0%) | 0 (0%) |

## 4. Discussion

The aim of the present study was to assess the influence of a constant resistance video-based engagement paradigm for pedalling on cadence control, as well as self-reported outcome measures including motivation and enjoyment in stroke patients and a contrasting cohort of healthy younger adults. A secondary aim was to compare the findings to cases pedalling with performance feedback data only and baseline pedalling (i.e., no feedback or engagement). The provision of performance feedback or video-based engagement during low-speed pedalling in stroke patients improved pedalling performance to a similar degree, as demonstrated by a lower speed deviation from the target and CV. While the capacity of stroke patients to maintain a fixed pedalling speed was improved with feedback and engagement, this occurred to a lesser extent at high cadence. As the ventilatory threshold is approached, the ability to meet the oxygen demands of the body is reduced, particularly that of the concentric and eccentric activations of the synergistic agonist and antagonist muscles that generate lower limb flexion and extension joint torques. Neuroplastic adaptations to stroke, altered motor commands, changes in motor performance, and cognitive decline are known to occur in stroke [28] and may have contributed to reduced capacity to maintain target pedalling cadence relative to the healthy adult subjects.



External audiovisual stimuli during video-mediated pedalling, and the enjoyment of performing exercise with feedback and engagement, may have provided a dissociation from the physiological cues associated with exertion during pedalling including muscle pain and heavy breathing. This is reflected in findings of an increase in peak pedalling speed for a given blood lactate level while watching a prerecorded video during pedalling, improvements in pedalling speed and distance, higher intensity exercise while cycling with video assistance, and no change in self-perceived exertion relative to pedalling without video stimulus [19,20]. However, altered brain activation, volitional motor commands, and pedalling performance following stroke are associated with abnormal muscle activation patterns, kinetics, and kinematics during pedalling [28]. As a result, the task of maintaining a constant cadence during pedalling while focusing on performance feedback data and video-based engagement is likely to be physically and cognitively more challenging in stroke patients compared to healthy adults. This may ultimately limit the capacity of stroke patients to dissociate themselves from the physical exertion of pedalling with feedback and engagement relative to healthy individuals, particularly for tasks requiring high exertion.

Enjoyment and efficacy are important factors in the adoption and long-term sustenance of exercise-based therapies [29,30], particularly in overcoming motivation barriers that prevent sedentary adults with functional deficits from being physically active. While this study is a proof of concept, and further long-term data are required to quantify the influence of feedback and engagement during pedalling on motor control, balance, and gait in rehabilitation, our preliminary data indicate high levels of enjoyment and self-reported motivation with exercise performance feedback and video-based engagement during pedalling for both stroke patients and healthy young adults. For instance, 92.3% of stroke patients either enjoyed or really enjoyed the video engagement regime, with 92.3% stating that video-based engagement would help motivate them to adhere to a routine of pedalling-based exercise therapy. Participants in a variety of other exercise monitoring studies have reported greater enjoyment in physical exercise when aided with video [29–31], music [17,20,32], gaming [33,34], or immersive virtual reality [35,36]. A shift from a focus on internal sensations to an external attentional focus, via external distracting stimuli, has been shown to correlate with a decrease in perceived exercise exertion and an increase in exercise enjoyment and performance for moderate intensity levels in adults [37]. However, further research ought to be performed to evaluate the effectiveness of engagement and feedback during progressive resistance training, which is a widely accepted rehabilitation approach to improve strength and function following injury or neuromuscular impairment [38].

This study has limitations that should be considered when interpreting the results. First, this was a single-centre study with a small heterogeneous sample of stroke patients, and the findings may be different in larger cohorts or patients with different levels of neuromuscular impairment. Second, some of the significant mean differences in cadence observed in this study between test conditions were small, and these data ought to be interpreted with caution. Third, the two cohorts assessed were not aged-matched, and so the influence of age on pedalling performance is a confounding variable. Nonetheless, the results provide broad preliminary evidence for the effectiveness of video-based engagement on pedalling performance in these two contrasting groups. Fifth, the duration of the pedalling exercise was chosen to minimise physical exertion and fatigue in stroke patients, and the results may vary with extended pedalling duration. In addition, the low and high speeds employed, which were selected in consultation with a stroke rehabilitation therapist, represent typical cadences used in stroke therapy, and deviation from these may ultimately influence pedalling control performance and engagement. Finally, this study employed video-based engagement, and while other strategies such as immersive gaming may also provide high levels of engagement and motivation, our approach may represent a less cognitively demanding activity suitable for longer-duration rehabilitation bouts, particularly in those that have neuromuscular impairment.

## 5. Conclusions

This proof-of-concept study showed that by providing feedback of cadence performance or video-based engagement, stroke patients were able to more effectively maintain a target cadence during stationary pedalling at a low target cadence. High levels of self-reported enjoyment during pedalling with the provision of performance feedback as well as video-based engagement indicate that these pedalling strategies may ultimately contribute to improved patient motivation and compliance in the exercise therapy or rehabilitation setting. The findings are applicable to the development of targeted therapy in the treatment of stroke and other neuromuscular conditions.

**Supplementary Materials:** The following supporting information can be downloaded at: https://www.mdpi.com/article/10.3390/app12147281/s1, Figure S1: Effect size data for absolute cadence deviation and coefficient of variation. The data shown are comparisons between the baseline pedalling and pedalling with feedback, the baseline pedalling and pedalling with video-based engagement, as well as comparisons between the pedalling with feedback and pedalling with video-based engagement, both for healthy adults and stroke patients.

**Author Contributions:** Conceptualization, T.W. and M.S.; methodology, M.S., T.W. and D.A.; software, M.S.; validation, M.S.; formal analysis, M.S. and D.A.; investigation, M.S., T.W. and D.A.; resources, T.W.; data curation, M.S.; writing—original draft preparation, M.S., T.W. and D.A.; writing—review and editing, D.A. and T.W.; visualization, M.S.; supervision, D.A. and T.W.; project administration, M.S. All authors have read and agreed to the published version of the manuscript.

**Funding:** This research received no external funding.

**Institutional Review Board Statement:** The study was conducted according to the guidelines of the Declaration of Helsinki and approved by the Ethics Committee of Western Health (HREC/16/WH/31, HREC 2015.308).

**Informed Consent Statement:** Written informed consent was obtained from all participants.

**Conflicts of Interest:** The authors declare no conflict of interest.

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
