# Peer review of "Cadence Feedback and Video-Based Engagement Improves Motivation and Performance during Pedalling in Stroke Patients"

_applsci, doi:10.3390/app12147281_

Round 1

Reviewer 1 Report

The article addresses an important issue: in fact the use of pedals in motor rehabilitation is common, but the cyclical and monotonous movement, accompanied by effort, often leads to demotivation and a decrease in pedalling cadence.

The authors propose a device that allows the maintenance of the cadence and the motivation (therefore, the exercise time).

The article is well structured, the discussion and the conclusion are clear and the limitations of the study are indicated.

However, the experimental methodology is only partially adequate: the main problem is that the control group is completely different in age from the experimental group.

The authors should justify this choice.

It would also be important to add:

1. A diagram of the connection pedal-control-video

2. Justify why the use of video and not a game (increases interactivity)

3. the calculation of the effect size

Author Response

Please see the attached response to reviewer comments document

Reviewer 2 Report

Dear Authors

I reviewed the topic "Cadence Feedback and Video-Based Engagement Improves Motivation and Performance During Pedaling in Stroke Patients". The topic of this research is interesting and of importance. I think the authors must have put a lot of effort into taking an experiment and into writing the manuscript. I think it will be a better thesis if only a few minor revisions presented below are revised.

Minor points:

Q1: Statistical symbol 'p' in abstract and text should be written in italics.

Q2: The source of the equipment or apparatus must be entered on Line 110.

Q3: It should explain why the pedaling speed of the bicycle was selected only at 60 rpm or 100 rpm on Line 133.

Q4: The reliability of the 'self-reported feedback survey' used as a survey method in this study should be presented on Line 141.

Q5: The method of calculating the samples and whether the sampling is normally distributed should be presented on Line 150.

Q6: As presented in lines 177 to 183, namely, “The dependent variables were absolute cadence deviation from target cadence (RPM), and coefficients of variation (%) of the cadence deviation. The independent variables were intervention type (baseline pedaling, pedaling with performance) feedback, and pedaling with video-based engagement), low and high pedaling speed (60 RPM and 100 RPM, respectively ), and participant type (healthy control or stroke patient). Significant differences in intervention type (Int), pedaling cadance (Cad), subject type (Sub) and their interactions are given, specifically, Int*Cad, Int*Sub, Cad*Sub, and Int*Cad*Sub." It should be sent as a footnote at the bottom of the table.

Q7: What does '57' in line 218 mean?

Q8: Authors must provide the full name of 'VR' on line 287 and use the abbreviation thereafter. The same is true for other cases.

Q9: References need to be presented in accordance with the MDPI format.

I hope my review helped you improve your manuscript.

Best regards,

Author Response

(The authors gave the same response as above.)
